# SERA: Soft-Verified Efficient Repository Agents

**Ethan Shen** [1 2]  **Daniel Tormoen** [1]  **Saurabh Shah** [1]  **Ali Farhadi** [1 2]  **Tim Dettmers** [1 3]

## Abstract

Open-weight coding agents should hold a fundamental advantage over closed-source systems because they can specialize to private codebases, encoding repository-specific information directly in their weights. Yet the cost and complexity of training has kept this advantage theoretical until now. We present **Soft-Verified Efficient Repository Agents (SERA)**, an efficient method for training coding agents that enables the rapid and cheap creation of agents specialized to private codebases. Using **Soft Verified Generation (SVG)**, we generate thousands of trajectories from any code repository, without requiring unit tests. Beyond repository specialization, we apply **SVG** to a larger corpus of codebases, generating 200,000+ synthetic trajectories. Using only supervised finetuning (SFT), **SERA** achieves leading results among fully open-source (open data, method, code) models while matching the performance of open-weight models like Devstral-Small-2. Creating **SERA** models is 26x cheaper than reinforcement learning and 57x cheaper than previous synthetic data methods to reach equivalent performance. We use our dataset to provide detailed analysis of scaling laws, ablations, and confounding factors for training coding agents. Overall, we believe our work will greatly accelerate research on open coding agents and showcase the advantage of open-source models that can adapt to private codebases.
Code: https://github.com/allenai/SERA
Blog: https://allenai.org/blog/open-coding-agents
Data: https://huggingface.co/collections/allenai/open-coding-agents

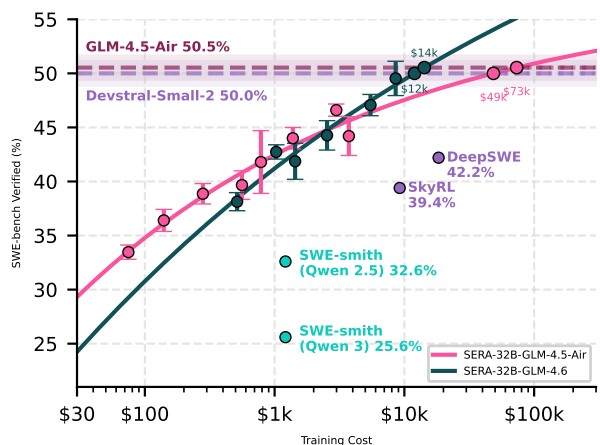

*Figure 1.* Scaling and cost comparison of coding agent training approaches using self-hosted vLLM inference. Horizontal lines indicate the cost at which our scaling law predicts matching Devstral-Small-2 and GLM-4.5-Air performance.

[1]Allen Institute of Artificial Intelligence, Seattle, Washington, USA [2]Paul G. Allen School of Computer Science and Engineering, University of Washington, Seattle, Washington, USA [3]Machine Learning Department, Carnegie Mellon University, Pittsburgh, Pennsylvania, USA. Correspondence to: Ethan Shen <ethans03@cs.washington.edu>, Tim Dettmers <dettmers@cmu.edu>.

*Proceedings of the $43^{rd}$ International Conference on Machine Learning*, Seoul, South Korea. PMLR 306, 2026. Copyright 2026 by the author(s).

## 1. Introduction

Coding agents have become central to software development and are increasingly applied to tasks beyond traditional engineering. While, closed-source coding agents are more powerful, open-weight models should hold a fundamental advantage in many applications because they can be specialized to private codebases, allowing them to learn repository-specific patterns, conventions, and domain knowledge. Despite this clear opportunity, the cost and complexity of training open-weight coding agents has kept this advantage theoretical. In this work, we show it is now practical.

Our method trains a 32B coding agent with simple supervised finetuning, achieving state-of-the-art open-source results at 40 GPU days ($2000) and matching strong open-weight models like Devstral-Small-2 at a budget of $9000. When specializing to a particular codebase, our pipeline can equal or exceed teacher model performance at $1300.

Training coding agents traditionally requires either reinforcement learning or complex synthetic data pipelines, demanding resources beyond what most teams can provide. Reinforcement learning requires sandboxed environments, distributed training infrastructure, and rollout orchestration. The complexity of this infrastructure is reflected in teams

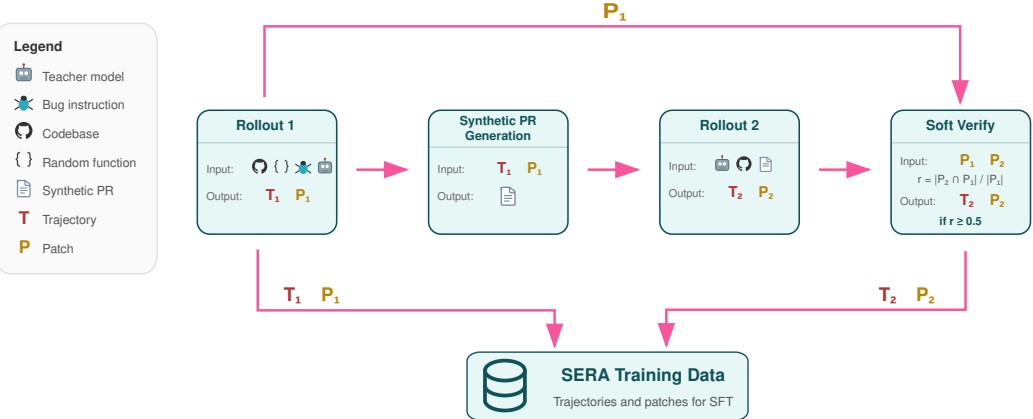

*Figure 2.* **Overview of SVG (Soft Verified Generation).** In the first rollout, a teacher model is prompted to make a change starting from a randomly selected function, producing a trajectory and patch. This trajectory is converted into a synthetic pull request. In the second rollout, the teacher attempts to reproduce the patch given only the PR description. Soft verification compares the two patches using line-level recall. We use $r \geq 0.5$ as an example threshold.

that average 12+ authors (Cao et al., 2025; Luo et al., 2025; Wei et al., 2025; Da et al., 2025). Synthetic data approaches like SWE-smith (Yang et al., 2025) require setting up test environments, generating valid bugs, and verifying bugs through test suites. These barriers have concentrated coding agent development in well-resourced industry labs and larger teams at academic institutions.

We found that much of the complexity in prior pipelines is unnecessary. Firstly, soft verification, where patches are checked by partial line-by-line matching rather than executed test suites, produces training data of equal quality to full test-based verification. At the scales we test, the degree of verification has minimal effect on downstream performance, removing the need for test infrastructure entirely and enabling data generation from any repository. Secondly, we observe that models prompted to fix these vague instructions often produce changes such as refactoring that are more representative of real world tasks than just bug fixes. Rather than requiring bug-focused data, we find that this general coding data is equally effective for performance and can be produced by prompting a model on any repository in its initial state. Together, these findings mean that generating high-quality training data requires neither test infrastructure nor complex bug-injection pipelines.

The resulting cost reduction and data abundance enable repository specialization. We show that open-weight models specialized to a codebase can match or exceed the performance of the teacher used to generate training data. This is intuitive: a student model with repository-specific knowledge encoded in its weights can outperform a teacher that only accesses the codebase through its context window. The advantage this creates extends beyond privacy. Even organizations willing to share code wait months until the next training run of a frontier model includes their data. LoRA

adapter options exist, but are often too costly for large-scale deployments. Open-weight specialization allows practitioners to generate data from their repositories, fine-tune, and deploy immediately, iterating as their codebase evolves. At low cost, any team can build and deploy a small specialized model that outperforms frontier systems on their codebase – an advantage that grows with codebase size and that frontier labs cannot close regardless of scale.

We introduce **SERA** (Soft-verified Efficient Repository Agents), a 32B coding agent that achieves 49.5%/54.2% on SWE-bench Verified when evaluated at 32K/64K context, state-of-the-art for fully open-source models. We exceed the performance of previous open-source solutions with a total cost of $2,000 for both data generation and training (40 GPU days). Our method, **SVG** (Soft Verified Generation; see Figure 2 for an overview), achieves equivalent performance to SkyRL at 26× lower cost and to SWE-smith at 57× lower cost when self-hosting inference via vLLM. Using the z.ai API, these advantages increase to 53× and 115× respectively. These efficiency factors are derived from our scaling laws that capture both per-sample savings and data quality gains (see Appendix E for detailed cost breakdowns). Effectively specializing to a single repository requires approximately 8,000 trajectories ($1,300). These trajectories are generated from randomly selected functions and contain no information about evaluation issues or their solutions. We validate our findings across multiple seeds and use scaling laws as robustness checks, adopting a methodology designed to ensure reported effects reflect genuine signal rather than noise. To further support open coding agent research, we provide extensive analyses covering ablations on data quality factors, model-specific pitfalls, and common confounding factors that have slowed progress in this area.

We release **SERA** along with all our code, 200,000 synthetic

coding agent trajectories, and Claude Code integration.

## 2. Background

### 2.1. SWE-bench

SWE-bench (Jimenez et al., 2023) is the standard benchmark for evaluating coding agents. Each task is derived from a real GitHub issue and pull request from popular Python repositories. Given an issue description, the agent must produce a patch that resolves the issue (previously failing tests pass and no previously passing tests break). Our primary evaluation, SWE-bench Verified, is a human-verified subset of SWE-Bench confirming each task is solvable.

### 2.2. Agent Scaffolds and Training Data

Coding agents operate through scaffolds that define available tools and environment interactions (Yang et al., 2024). A **rollout** is one complete execution of the agent on a task. The sequence of actions, observations, and reasoning produced during a rollout is called a **trajectory**. A **patch** specifies final additions and deletions to the codebase. Trajectories are the training data for coding agents.

### 2.3. Synthetic Data Generation

Synthetic data generation creates trajectories by having a teacher model solve synthetic tasks, then using those trajectories to train a student model. The standard approach, used by SWE-smith (Yang et al., 2025), generates data through bug injection: starting with a repository with passing tests, the pipeline injects bugs that cause tests to fail, has the teacher solve the issue, and verifies correctness by checking that tests pass again. This requires test infrastructure and execution environments for verification. A limitation of teacher-student distillation is that the student's performance is often bounded by the teacher's capability. At the frontier, where no stronger teacher exists, synthetic data may not suffice and reinforcement learning might be required.

### 2.4. Reinforcement Learning

Reinforcement learning trains coding agents by having them generate rollouts and learn from reward signals from whether tasks are solved. Unlike synthetic data generation, the model being trained is also the model producing rollouts. The advantage is that a strong model can continue to improve without being bounded by a teacher. The disadvantage is infrastructure complexity: online rollouts during training, sandboxed execution environments, and distributed systems for coordinating gradient updates. This motivated our focus on supervised methods.

### 2.5. Verification

Verification determines whether a generated trajectory is suitable for training. Traditional approaches use **unit test verification**: the patch must pass all relevant tests, confirming the synthetic bug was resolved. This ensures correctness but limits data generation to repositories with test coverage.

Our method introduces **soft verification**: instead of execut-

ing tests, we compare the generated patch against a reference patch using line-level recall. If the generated patch contains most or all of the lines from the reference patch, we consider it verified. This removes the need for test infrastructure and enables data generation from any repository. We describe the details in the following section.

## 3. Method

### 3.1. Soft Verified Generation (SVG)

The key intuition behind **SVG** is that errors in syntax, logic, and failed unit tests are only a subset of real world pull requests (PRs). Indeed, it is common for PRs to be more obscure, aimed at refactoring code, style requirements, or tweaking behavior.

In **SVG** (Figure 2), we rethink the criteria that define a valid synthetic PR. While traditional approaches focus synthetic issues on failed unit tests to ensure samples represent correct code, we broaden the definition of a PR to include any instruction that creates *some* desired change in a codebase $C$. The key insight is that a trajectory's value for training lies not in producing a fully correct patch, but in the skills it demonstrates: how to interpret an instruction, navigate a codebase, and translate intent into code.

**SVG** is composed of two rollouts. We use $T$ and $P$ to denote the trajectory and patch created by a rollout. In SVG, we use a teacher model $M$ to generate rollouts. In the first rollout, we prompt $M$ with a random function $\text{func}_i$ from codebase $C$ and a bug prompt $\text{bug}_j$ sampled from a set of 51 bug types $B$. This produces trajectory $T_1$ and patch $P_1$. We then convert $T_1$ into a synthetic PR $\text{synth\_PR}$ using a demonstration PR $PR$ sampled from SWE-Bench Verified. In the second rollout, $M$ is prompted with $\text{synth\_PR}$ and tasked to reproduce the original change, producing trajectory $T_2$ and patch $P_2$. Soft-verification compares $P_2$ against $P_1$ using line-level recall $r$. Together, these steps form **SVG**.

$$T_1, P_1 = M(\text{func}_i, \text{bug}_j, C) \qquad (1)$$

$$\text{synth\_PR} = M(T_1, PR) \qquad (2)$$

$$T_2, P_2 = M(\text{synth\_PR}, C) \qquad (3)$$

$$r = \frac{|P_2 \cap P_1|}{|P_1|} \qquad (4)$$

If $r = 1$, the trajectory is hard-verified; if $0 < r < 1$, soft-verified; if $r = 0$, unverified.

**Agent Workflow:** We use SWE-agent (Yang et al., 2024) for rollouts, which gives users the ability to adjust settings such as the tools available, length of tool outputs, and context history. To reduce confounding factors, we use SWE-agent in its vanilla state and only provide the agent with the ability to run tools for viewing, editing, submitting, and bash commands. Furthermore, we do not truncate context history or tool outputs at any point. While truncations are frequently done to avoid context window errors, we noticed that many works use slightly different heuristics, making it difficult to

compare performance. Additionally, we believe an important trait of coding agents is their ability to solve tasks while avoiding unnecessarily long tool calls and outputs.

**First Rollout:** The first rollout works as follows: We prompt the teacher model $M$ with "There is a $bug_j$ downstream of function $func_i$.", where $bug_j$ vaguely describes a bug type and $func_i$ is a random function in the codebase $C$. The function $func_i$ serves as an arbitrary starting point for the agent. We run the pipeline once for every function in $C$. Each $bug_j$ is randomly sampled from a list of 51 bug types $B$ and asks the model to fix issues ranging from state management to code clarity. We generate this list from papers that study bug distributions in software systems (Just et al., 2014; Widyasari et al., 2020). We intentionally leave the prompt vague to widen the range of acceptable changes and rollout up to 115 steps, although this limit is rarely reached.

Next, we ask $M$ to self-evaluate its fix. We accept the trajectory $T_1$ unless the teacher model decides it did not make an aligned change. In that case, we reject the trajectory and we perform another rollout with a newly sampled prompt until a valid change is made or a limit of three runs is reached. About 2% of rollouts are rejected and $< 1\%$ fail all three rollouts. We also discard trajectories that produce duplicate patches, though this is extremely rare. The patches $P_1$ from accepted trajectories are saved as ground truth (Equation 1).

**Synthetic PR:** Next, to create $synth\_PR$ to guide the second stage, we provide the teacher model $M$ with its first rollout $T_1$, which contains relevant reproduction scripts, execution traces, and the final software patch. Similar to SWE-smith (Yang et al., 2025), we also include a demonstration PR $PR$ sampled from SWE-Bench Verified (Jimenez et al., 2023). The teacher is then asked to write a new PR that follows the format of the demonstration PR (Equation 2).

**Second Rollout:** In the second rollout, we only use the synthetic PR $synth\_PR$ as input, with the goal of reproducing the initial patch (Equation 3). The trajectory $T_2$ is again capped at 115 steps, and the resulting patch $P_2$ is saved.

**Soft Verification:** We evaluate $P_2$ using recall against $P_1$ by assessing edits at a line-by-line granularity (Equation 4). If $P_2$ contains every change from $P_1$, then the line-level recall is $r = 1$ and the second rollout is hard-verified; if $0 < r < 1$ it is soft-verified; if $r = 0$, then it is unverified.

**Setup Details:** We use a suite of 121 codebases for data generation, which are a subset of the 128 codebases released by SWE-smith (Yang et al., 2025).[1] Each codebase $C$ is encapsulated inside of a docker container. We use GLM-4.5-Air (GLM-4.5 Team et al., 2025) as our teacher model $M$ for all experiments unless otherwise specified. GLM-4.5-Air has many advantages: it is relatively small, has full reasoning traces, and is performant. Pairing $SVG$'s generation effi-

ciency and GLM-4.5-Air's cost efficiency makes a robust scientific investigation of coding agent scaling possible.

Indeed, current data generation strategies for coding agents are bottlenecked by their reliance on closed-source teachers, whose API costs make studying scaling impractical and hamper statistical reliability, since evaluations across multiple random seeds becomes costly. Closed-source providers may also hide full reasoning traces, which are key to data quality (Appendix G), and APIs are prone to adjust model quality depending on demand. As an open-weight model, GLM-4.5-Air can be run locally, avoiding these issues. GLM-4.5-Air also strikes a balance between performance and model size: it provides Claude 3.7 Sonnet[2] level performance while being fully deployable on 8 H100s, or 4 H100s at a lower context length, or 2 H100s if quantized. This significantly reduces barriers for practitioners and researchers who want to train, use, and study coding agents at scale.

### 3.2. Training

We use Qwen 3-32B (Team et al., 2025b) as our primary base model over models like Qwen 2.5 (Team et al., 2024) due to Qwen 3-32B's stronger tool calling performance, which better reflects the improving capabilities of current and future base models. This mirrors choices from recent works (Sonwane et al., 2025b; Cao et al., 2025; Luo et al., 2025). We fully fine-tune up to Qwen 3's native context length of 32768 and train our models for 3 epochs using a learning rate of `1e-5` and weight decay of 0.01. We prioritize training on trajectories inherently shorter than 32768 tokens. To increase sample size, we truncate longer trajectories based on the ratio of trajectory steps within the context limit—we term this "truncation ratio". In Section 5.3, we explain why truncation must be done with caution.

## 4. Main Results

We primarily evaluate on SWE-bench Verified (Jimenez et al., 2023), comparing against recent open-weight coding agents including Nex-N1-8B (Team et al., 2025a), CWM (Copet et al., 2025), and models built on InternLM3-8B (Cai et al., 2024), as well as synthetic data and RL approaches (Table 1). We focus on three evaluation settings: (1) a head-to-head comparison that controls for the teacher model while comparing against other synthetic data methods, (2) a scaling law study that examines how our approach scales with data size and predicts when we reach certain performance thresholds, and (3) a benchmarking of how well $SERA$ can target specific codebases for training.

A key consideration in our evaluation methodology is controlling for evaluation context length. Context length significantly impacts memory footprint even among models of equal sizes. Doubling the context length often increases memory usage by nearly the same factor. We also observe

---

[1]We exclude 7 codebases that contain little to no Python code.

[2]https://www.anthropic.com/news/claude-3-7-sonnet

*Table 1.* SWE-bench Verified performance grouped by context length. Gray rows: open-weight models; white rows: fully open-source. **SERA** achieves state-of-the-art among fully open-source models at both 32K and 64K context. Standard deviations from 3 seeds.

| Method | Open Source | | | Base Model | Teacher | Context | Resolve Rate |
|---|---|---|---|---|---|---|---|
| | Code | Model | Data | | | | |
| SkyRL-8B | ✓ | ✓ | | Qwen 3-8B | — | 32K | 9.4% |
| Nex-N1-8B | ✓ | ✓ | | InternLM3-8B | — | 32K | 20.3% |
| **SERA**-8B-GA (Ours) | ✓ | ✓ | ✓ | Qwen 3-8B | GLM-4.5-Air | 32K | 31.7% $\pm$ 0.4% |
| **SERA**-8B (Ours) | ✓ | ✓ | ✓ | Qwen 3-8B | GLM-4.6 | 32K | 31.7% $\pm$ 0.9% |
| Qwen 3-32B | ✗ | ✓ | ✗ | 32B | — | 32K | 24.4% |
| SWE-smith | | | | Qwen 3-32B | Claude 3.7 | 32K | 25.6% $\pm$ 1.1% |
| SWE-smith | ✓ | ✓ | ✓ | Qwen 2.5-32B | Claude 3.7 | 32K | 32.6% |
| FrogBoss-32B | ✗ | ✓ | ✗ | Qwen 3-32B | Claude 4 Sonnet | 32K | 35.0% |
| GLM-4.7-Flash | ✗ | ✓ | ✗ | 30B | — | 32K | 37.3% $\pm$ 2.0% |
| DeepSWE | ✓ | ✓ | | Qwen 3-32B | — | 32K | 42.2% |
| Kimi-dev | ✗ | ✓ | ✗ | 72B | — | 32K | 48.6% |
| Devstral-Small-2 | ✗ | ✓ | ✗ | 24B | — | 32K | 50.0% $\pm$ 1.3% |
| GLM-4.5-Air | ✗ | ✓ | ✗ | 110B | — | 32K | 50.5% $\pm$ 1.3% |
| **SERA**-32B-GA (Ours) | ✓ | ✓ | ✓ | Qwen 3-32B | GLM-4.5-Air | 32K | 46.6% $\pm$ 0.7% |
| **SERA**-32B (Ours) | ✓ | ✓ | ✓ | Qwen 3-32B | GLM-4.6 | 32K | 49.5% $\pm$ 1.9% |
| GLM-4.7-Flash | ✗ | ✓ | ✗ | 30B | — | 64K | 39.7% $\pm$ 1.8% |
| SWE-Swiss | ✓ | ✓ | | Qwen 2.5-32B | — | 128K | 45.0% |
| Qwen 3-Coder-30B | ✗ | ✓ | ✗ | 30B | — | 256K | 51.6% |
| CWM | ✗ | ✓ | ✗ | 32B | — | 128K | 53.9% |
| FrogBoss-32B | ✗ | ✓ | ✗ | Qwen 3-32B | Claude 4 Sonnet | 64K | 54.6% |
| GLM-4.5-Air | ✗ | ✓ | ✗ | 110B | — | 64K | 57.4% $\pm$ 0.5% |
| Devstral-Small-2 | ✗ | ✓ | ✗ | 24B | — | 64K | 59.1% $\pm$ 1.1% |
| GLM-4.7-Flash | ✗ | ✓ | ✗ | 30B | — | 128K | 59.2% |
| Devstral-Small-2 | ✗ | ✓ | ✗ | 24B | — | 256K | 68.0% |
| **SERA**-32B-GA (Ours) | ✓ | ✓ | ✓ | Qwen 3-32B | GLM-4.5-Air | 64K | 51.7% $\pm$ 1.1% |
| **SERA**-32B (Ours) | ✓ | ✓ | ✓ | Qwen 3-32B | GLM-4.6 | 64K | 54.2% $\pm$ 1.4% |

that context length is a factor that strongly differentiates performance—methods evaluated at 64K or 128K context appear much stronger than those evaluated at 32K context, even when underlying model capabilities are similar. To ensure fair comparisons across deployment configurations, we explicitly report and control for context length in our leaderboard (Table 1). For every experiment, performance is averaged over three random seeds, which is essential for reliable conclusions given high variance in coding agent evaluations (detailed statistical analysis in Appendix I).

### 4.1. Controlled Comparisons

The goal of this section is to understand the differences between **SERA** and other synthetic data generation methods. Because data generation varies based on repository and teacher model, sample sizes can only be approximately matched, introducing some variance in final dataset sizes.

We also note an important methodological consideration

regarding context management during evaluation. Some agent frameworks employ optimizations such as retaining only the last few tool calls in context rather than the full trajectory history. While this compression allows models to appear effective at longer context lengths, it introduces a significant confounding factor when evaluating model capability. Furthermore, such optimizations cause key-value cache invalidation during inference, which is prohibitively expensive for practical deployment. For fair comparison, we evaluate all methods using full context retention, ensuring that reported context lengths accurately reflect the information available to the model.

We compare against SWE-smith and BugPilot using hard-verified trajectories from the second rollout. This ensures that our training data distribution mimics that of other setups (i.e. synthetic issue descriptions and working code). Table 2 shows that with the same teacher and sample size as SWE-

*Table 2.* Controlled comparison with matched teacher models and sample sizes on Qwen 3-32B. **SERA** outperforms SWE-smith (+4.7%) and matches BugPilot despite 10% fewer samples. See Appendix B for additional comparisons.

| Method | SWE-smith | **SERA** | BugPilot | **SERA** |
|---|---|---|---|---|
| Base model | Qwen 3-32B | Qwen 3-32B | Qwen 3-32B | Qwen 3-32B |
| Teacher | Claude 3.7 | Claude 3.7 | Claude 4 Sonnet | Claude 4 Sonnet |
| Eval context size | 32K | 32K | 64K | 64K |
| Sample size | 4776 | 4776 | 5819 | 5319 |
| SWE-bench Verified | 25.27% ± 0.61% | 30.00% ± 1.41% | 49.87% | 48.53% ± 0.31% |

smith, **SERA** yields better performance. Appendix B further shows that SWE-smith is optimized for Qwen 2.5 and that GLM-4.5-Air is the optimal teacher model over 3.7 Sonnet.

We also evaluate at 64K context against BugPilot BaseMix (Sonwane et al., 2025a), the base mixture in BugPilot's training data, which combines real and synthetic issues from R2EGym and SWE-smith with Claude 4 Sonnet as teacher. BugPilot's data is not public, so we compare to its BaseMix performance reported in the paper because its sample size is closest to our largest Claude 4 Sonnet run. Still, our train set contains approximately 10% fewer samples. Despite this, in a head-to-head comparison, our results nearly match BugPilot's performance.

These results demonstrate that the data quality of our approach is high. Controlling for the teacher model, **SERA** matches unit-test based real and synthetic approaches.

### 4.2. Scaling Experiments

**SERA** simplifies the process of generating massive amounts of coding data by circumventing the need to introduce synthetic bugs into codebases and validate them with unit tests. We take advantage of this property to generate three large-scale datasets from the codebases described in Section 3, using GLM-4.5-Air and GLM-4.6 as teachers (Figure 1).

- **Sera-4.5A-Lite** is generated by running our data generation pipeline once for every function across all 121 codebases using GLM-4.5-Air as the teacher. This results in approximately 36,000 $T_1$ and 36,000 $T_2$ trajectories.
- **Sera-4.5A-Full** is a superset of **Sera-4.5A-Lite**. We continue our generation from **Sera-4.5A-Lite**, looping through every $func_i$ up to three total times. Each time, a new bug prompt is sampled for the first rollout. This ensures that every trajectory is unique even for the same $func_i$. We stop generation after several days, reaching a total of 70,000 $T_1$ and 70,000 $T_2$ trajectories.
- **Sera-4.6-Lite** mimics the setup of **Sera-4.5A-Lite**, but uses GLM-4.6 as the teacher model. We generate another 36,000 $T_1$ and 36,000 $T_2$ trajectories for **Sera-4.6-Lite**.

Combined, our datasets contain over 200,000 trajectories, resulting in the largest open-source dataset for coding agents

to date. We separate these trajectories by teacher model and rollout stage. We independently scale both $T_1$ trajectories and $T_2$ trajectories until a truncation ratio of 0.88 is reached. We choose to scale $r = 0$ (completely unverified) trajectories since they have the highest data count. Our decision was influenced by experiments in Section 5.1, which indicate that completely unverified $T_2$ rollouts are of equal or better quality than any verified rollouts.

Using **Sera-4.6-Lite**, we train SERA-32B and set a new state-of-the-art on SWE-Bench Verified for fully open-source 32B models evaluated at 32K context, with open-weight models like Devstral-Small-2-24B and larger models like GLM-4.5-Air well within uncertainty bounds of one standard deviation. Evaluating at 64K context, SERA-32B again sets a state-of-the-art among fully open-source models, matching open-weight models such as FrogBoss-32B and only outperformed by Devstral-Small-2-24B (Rastogi et al., 2025) among models with similar parameter counts. Importantly, unlike these models, SERA-32B was not trained past 32K tokens or with reinforcement learning, which disadvantage it at longer contexts. Still, SERA-32B performs very well and does not appear to have saturated yet.

We also train SERA-32B-GA using **Sera-4.5A-Lite**. While SERA-32B-GA lags behind SERA-32B, it still outperforms all otherr fully open-source models at 32K and 64K context. Interestingly, we find that SERA-32B-GA is able to match SERA-32B at low and intermediate sample sizes, after which SERA-32B-GA's performance saturates. This suggests that the benefits of strong teacher models primarily emerge in high compute regimes. For researchers and practitioners, this means that it may be optimal to use a weaker teacher depending on final performance goals and budget. Figure 1 highlights this crossover point where the scaling curves for SERA-32B-GA and SERA-32B intersect. Overall, **SERA** is up to 53× cheaper than reinforcement learning and 115× cheaper than previous synthetic data methods to reach equivalent performance when using the z.ai API; we provide detailed cost calculations in Appendix E.

### 4.3. Repository Specialization

**SERA** is the first synthetic data generation strategy that operates totally independent of a repository's unit tests. This

allows users to rapidly specialize a base model to any downstream codebase, including private repositories. To emulate this scenario, we use **SERA** to generate data from the three largest repositories in SWE-Bench Verified: Django, Sympy, and Sphinx. Crucially, our synthetic training data is generated independently of the evaluation instances, containing no information about their solutions. These repositories represent 231 (46.2%), 75 (15.0%), and 44 (8.8%) of the 500 instances in SWE-Bench Verified, respectively.

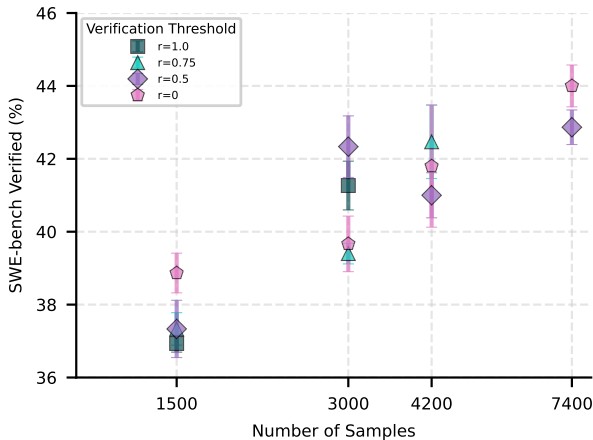

*Figure 4.* Verification analysis comparing soft and hard verification. All thresholds achieve similar performance at each scale, indicating strict verification provides no benefit over soft or unverified data.

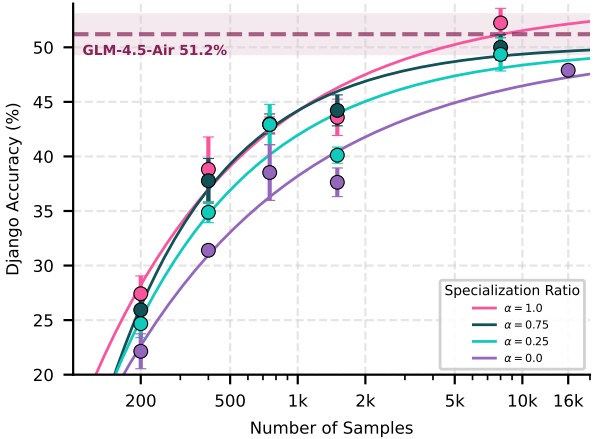

*Figure 3.* Scaling law for repository specialization on Django. The specialization ratio $\alpha$ denotes the fraction of Django-specific data in the training mixture. Full specialization ($\alpha = 1.0$) matches teacher performance at $\sim$8,000 samples; performance increases with specialized data ratio.

**Initial Data Generation:** Because every SWE-Bench instance is sourced from a unique commit, the set of instances from each repository will span multiple years. To account for this, we identify the earliest commit and latest commit in SWE-Bench Verified from each repository and generate data from five equally spaced commits in that period. While some functions are repeated across commits, each commit presents the codebase in a different context, which ensures trajectories in the first and second rollout remain unique. Aggregating across commits, we obtain between 46000 and 54000 trajectories for each repository combined across both rollouts. We decide to train on both rollouts to increase sample size since the majority of the generated trajectories exceed 32768 tokens. We investigate the effects of mixing rollouts in Appendix H. Due to compute constraints, we train on 8000 trajectories per repository rather than the full dataset; however, we release all generated trajectories to enable future research to explore larger-scale specialization.

**Data Verification and Filtering:** We soft-verify $\mathbf{T_2}$ rollouts with a verification threshold of 0.5. We cap $\mathbf{T_1}$ rollouts based on patch size and observations length. This selects against $\mathbf{T_1}$ rollouts that over-edit or make excessively long tool calls, a tendency that can quickly use up context. We find that this filtering significantly improves specialization

performance, which we further investigate in Appendix F.

Finally, for each repository, we train on 3,000 soft-verified $\mathbf{T_2}$ rollouts and 5,000 filtered $\mathbf{T_1}$ rollouts. We note that these specific proportions were chosen based on preliminary experiments within our compute budget; a more systematic exploration of the optimal mixture would be valuable future work. In this setup, we match or exceed the teacher model GLM-4.5-Air on Django and Sympy instances and also outperform Devstral-Small-2-24B (SoTA $\leq$32B parameters), while nearly matching their performances on Sphinx (Table 3). We note that at 64K evaluation context, **SERA** underperforms baselines because we train only at 32K context while these models are trained at 64K or longer; see Appendix C for 64K results. These results highlight that given the right data, it is possible to match and even exceed state-of-the-art performance on specifically targeted repositories.

**Specialization Scaling Law:** To understand how data composition affects specialization efficiency, we fit scaling laws across different mixtures of Django-specific and general coding data (Figure 3). The specialization ratio $\alpha$ is the fraction of repository-specific data in the training mixture. At $\alpha = 1.0$ (pure Django data), the model matches teacher performance (GLM-4.5-Air at 51.2%) with only 8,000 samples. In contrast, $\alpha = 0.0$ (pure general data) is unable to reach equivalent performance even at 16,000 samples. Intermediate mixtures ($\alpha = 0.75$, $\alpha = 0.25$) show increasing asymptotic performance as the proportion of specialized data increases. This indicates that the ratio of target codebase data is the most important factor when specializing.

## 5. Ablations and Analysis

We conduct comprehensive data ablations studying design choices in **SERA**. We focus on the impacts of verification and truncation. In the Appendix, we further explore spe-

*Table 3.* Repository specialization at 32K context. Student models trained on 8,000 synthetic trajectories match or exceed the teacher (GLM-4.5-Air) on Django and Sympy. Results averaged over 3 seeds; see Table 7 for 64K results.

| Model | Django (231) | Sympy (75) | Sphinx (44) |
|---|---|---|---|
| SERA-32B-Django | **52.23% ± 1.64%** | - | - |
| SERA-32B-Sympy | - | **51.11% ± 1.54%** | - |
| SERA-32B-Sphinx | - | - | 37.14% ± 6.95% |
| GLM-4.5-Air | 51.20% ± 1.80% | 48.89% ± 3.08% | **43.51% ± 0.58%** |
| Devstral-Small-2-24B | 51.30% ± 1.72% | 47.56% ± 4.68% | 38.95% ± 4.24% |

cialization, filtering, rollout mixing, teacher choice, and evaluation uncertainty. For data, we use **Sera-4.5A-Lite**.

### 5.1. Verification

Traditional synthetic data methods require unit test verification to ensure correctness. We test whether this is necessary by comparing four verification thresholds ($r = 0.0, 0.25, 0.75, 1.0$) on $T_2$ trajectories (Figure 4). We only train on full trajectories without truncation to ensure fair comparison. If verification was essential, we would expect performance to increase with stricter thresholds. Instead, all thresholds perform similarly up to 7,400 samples—even completely unverified trajectories match hard-verified ones. This suggests verification is not essential for high-quality coding data, consistent with findings in other reasoning domains (Chandra et al., 2026). We hypothesize that even incorrect trajectories contain important skills like navigating codebases and translating intent into code.

### 5.2. Line-Level Recall

Verification and reinforcement learning may become necessary once a model saturates on skills that unverified trajectories teach. In this data regime, an important question is whether line-level recall can be used to measure code correctness. We find that the answer is a resounding yes. Across 1,036 SWE-Bench Verified evaluations, comprised of over 500,000 individual tasks, we find an extremely high correlation between line-level recall and unit test resolution, illustrated in Figure 5. This suggests that line-level recall can serve as a continuous proxy for code correctness, while having the added benefits of being cheaper, quicker to run, and generalizable to any codebase.

### 5.3. Truncation

Truncation is common practice in coding agent training because approximately 25% of trajectories will exceed base model context windows. Current methods typically slice long trajectories arbitrarily to fit within context limits. We

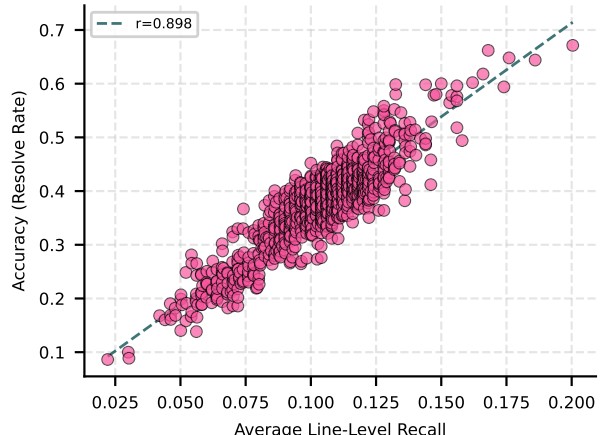

*Figure 5.* Scatterplot comparison of issue resolution rates and line-level recall across 1,036 SWE-Bench Verified evaluations using a mixture of open and closed weight models.

find that this random truncation is suboptimal and that the fraction of trajectory steps retained ("truncation ratio") strongly predicts data quality.

We compare three strategies: use only trajectories that fit in 32K, ordered truncation (selecting sliced trajectories with the highest truncation ratios), and randomly selecting sliced trajectories. As shown in Table 4, ordered truncation outperforms random truncation by 5.5% and even exceeds fully-fitting trajectories by 2.4%. This finding informs our scaling experiments, where we extend datasets by including truncated trajectories only above a ratio of 0.88. We provide detailed analysis across truncation ratios in Appendix D.

## 6. Related Work

**Synthetic Data Generation:** SWE-smith (Yang et al., 2025) is the most closely related work, generating training data through bug injection by breaking existing tests in Python codebases and producing 50K instances from 128 repositories. BugPilot (Sonwane et al., 2025a) takes a different approach, synthesizing bugs by instructing agents to add features and using unintentional test failures as training data. Other approaches include SWE-Synth (Pham et al., 2025), which simulates debugging workflows to produce bug-fix

*Table 4.* Ordered truncation (highest ratios first) outperforms random by 5.5%. All use 3,000 $T_1$ trajectories.

| Fully Fit | Ordered | Random |
|---|---|---|
| 40.60% ± 0.69% | **43.00% ± 1.93%** | 37.47% ± 0.50% |

pairs, and SWE-Mirror (Wang et al., 2025), which distills real Github issues into configured environments.

**Reinforcement Learning:** SkyRL-Agent (Cao et al., 2025) and DeepSWE (Luo et al., 2025) train coding agents through online rollouts with reward signals based on test execution. SWE-RL (Wei et al., 2025) applies RL on software evolution data using rule-based rewards. These approaches require substantial infrastructure for training and execution, which motivated our focus on SFT.

**Agent Scaffolds:** SWE-agent (Yang et al., 2024) introduced a custom agent-computer interface with tools for viewing, editing, and bash execution; we use SWE-agent for all experiments. OpenHands (Wang et al., 2024) provides a modular platform with standardized tool interfaces and sandboxed execution. Agentless (Xia et al., 2024) demonstrates that good performance can be achieved through a simpler three-phase approach without complex scaffolding.

**Datasets and Environments:** SWE-Gym (Pan et al., 2024) provides the a dedicated training environment with over 2,400 Python task instances. R2E-Gym (Jain et al., 2025) scales environment curation through test generation and commit back-translation. Terminal-Bench (Merrill et al., 2026) provides a comprehensive evaluation covering diverse terminal-based tasks beyond repository-level code editing.

## 7. Limitations

In Appendix I, we discuss the robustness of coding agent evaluations. We design our experiments to be as robust as possible, but may not have strong signal for all claims. We also only evaluate on SWE-bench Verified, limiting our understanding of broader generalization.

We show different verification levels, including no verification, perform equally. One explanation is that early gains depend on learning skills like converting intentions into code edits and navigating codebases rather than code correctness. However, once a model saturates on these aspects, verification and reinforcement learning may be necessary.

Finally, we demonstrate specialization on Django, Sympy, and Sphinx because they have evaluation data; we have not verified specialization on truly private codebases.

## 8. Conclusion

In conclusion, we present **SERA**, a suite of leading open-source coding agents trained with **SVG**, a novel data generation strategy that verifies code patches using line-level recall instead of unit tests. This allows data to be generated from *any* repository, enabling rapid and cheap specialization of coding agents to private codebases. We scale **SVG** to generate over 200,000 trajectories from a broad group of repositories, resulting in state-of-the-art performance on SWE-Bench Verified at a fraction of the cost of reinforcement learning and other synthetic data approaches. Finally, we conduct extensive ablations on our dataset and find that verification is not always necessary for coding agent training—instead other heuristics like truncation ratio can have a larger impact. We believe **SERA** will have significant impact on the research community by enabling research on coding agents without requiring large resources or complicated systems, while also demonstrating the strong potential of specialized agents for coding.

## Acknowledgments

This research was supported by an AI2050 Early Career Fellowship and by a Laude Institute Slingshot. We thank Taira Anderson, Caroline Wu, Johann Dahm, Sam Skjonsberg, David Albright, Kyle Wiggers, Hanna Hajishirzi, Ranjay Krishna, Crystal Nam, and the Beaker Team for their feedback and support.

## Impact Statement

This paper presents work whose goal is to advance the field of machine learning. There are many potential societal consequences of our work, none of which we feel must be specifically highlighted here.

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

# A. Scaling Law and Data Points

We fit a power law to our cost-performance data to predict how SERA scales with additional investment. The scaling law takes the form:

$$y = c - a \cdot x^{-b}, \tag{5}$$

where $y$ is the SWE-bench Verified resolve rate (%), $x$ is the total training cost in thousands of dollars (including both data generation and training), $c$ is the asymptotic performance ceiling as cost approaches infinity, $a$ is a scaling coefficient controlling how far below the asymptote performance begins, and $b$ is the power law exponent governing the rate of diminishing returns. The curve is fitted separately for each cost regime (vLLM self-hosting at \$0.187/sample and z.ai API at \$0.092/sample), yielding different $(c, a, b)$ parameters since the same number of samples maps to different costs.

To predict the cost of matching a baseline system, we solve Equation 5 for $x$ at the target performance level $y^*$:

$$x^* = \left( \frac{a}{c - y^*} \right)^{1/b}. \tag{6}$$

For example, Devstral-Small-2 achieves 50.0% and GLM-4.5-Air achieves 50.5% on SWE-bench Verified. Solving for these targets yields predicted costs of \$7K (z.ai API) or \$15K (vLLM) to match Devstral-Small-2, and \$9K (z.ai API) or \$19K (vLLM) to match GLM-4.5-Air. The fitted asymptote is approximately 70%, suggesting substantial headroom remains if data quantity is scaled further, though we note this extrapolation is uncertain as it extends well beyond our observed data range.

Table 5 provides the exact data points underlying the scaling law in Figure 1. All experiments use Qwen 3-32B as the base model trained on SERA data generated with GLM-4.5-Air as the teacher, evaluated on SWE-bench Verified at 32K context length. Each condition is evaluated over 3 random seeds. We report these values to enable other researchers to directly compare against our scaling curve without needing to read approximate values from the plot.

*Table 5.* Exact scaling law data points for Figure 1. Performance is SWE-bench Verified resolve rate (%). Costs include both data generation and training. Per-sample cost is \$0.187 for vLLM self-hosting and \$0.092 for the z.ai API.

| Samples | Seed 1 | Seed 2 | Seed 3 | Mean (%) | Std (%) | Cost (vLLM) | Cost (z.ai) |
|--------:|-------:|-------:|-------:|---------:|--------:|------------:|------------:|
| 400 | 34.40 | 33.00 | 33.00 | 33.47 | 0.81 | \$75 | \$37 |
| 750 | 36.80 | 35.00 | 37.40 | 36.40 | 1.25 | \$140 | \$69 |
| 1,500 | 38.20 | 40.20 | 38.20 | 38.87 | 1.15 | \$280 | \$138 |
| 3,000 | 40.60 | 37.80 | 40.60 | 39.67 | 1.62 | \$560 | \$275 |
| 4,200 | 40.60 | 45.80 | 39.00 | 41.80 | 3.56 | \$784 | \$386 |
| 7,400 | 43.20 | 45.40 | 43.40 | 44.00 | 1.22 | \$1,382 | \$679 |
| 16,000 | 47.00 | 47.00 | 45.80 | 46.60 | 0.69 | \$2,987 | \$1,469 |

# B. Additional Baseline Comparisons

Table 6 provides additional baseline comparisons that complement the main results in Table 2. We train Qwen 2.5-32B on SWE-smith, which performs much better than when transferred to Qwen 3. This suggests that SWE-smith is optimized for Qwen 2.5-32B. We also include the **SERA** result using GLM-4.5-Air as a teacher, which shows the substantial performance improvement from using a stronger teacher model compared to Claude 3.7.

# C. Specialization Results at 64K Context

Table 7 presents specialization results evaluated at 64K context length. Because **SERA** models are trained at 32K context while competing models like Devstral-Small-2 are trained at 64K or longer contexts, **SERA** underperforms at 64K evaluation despite matching or exceeding these baselines at 32K (Table 3). This context length mismatch explains the performance gap: our models have not learned to effectively utilize the additional context available at 64K tokens.

*Table 6.* Additional baseline comparisons. SWE-smith with Qwen 2.5-32B shows the method was optimized for this model family. **SERA** with GLM-4.5-Air demonstrates the benefit of stronger teacher models.

| Method | SWE-smith | **SERA** |
|---|---|---|
| Base model | Qwen 2.5-32B | Qwen 3-32B |
| Teacher | Claude 3.7 | GLM-4.5-Air |
| Eval context size | 32K | 32K |
| Sample size | 6402 | 4933 |
| SWE-bench Verified | 32.60% | 38.47% $\pm$ 1.01% |

*Table 7.* Specialization results at 64K context. Fine-tuned **SERA** models underperform baselines at 64K because they are trained at 32K context, while Devstral-Small-2 is trained at longer contexts. Results averaged over three seeds.

| Model | Django (231) | Sympy (75) | Sphinx (44) |
|---|---|---|---|
| Qwen 3-32B-Django | 56.56% $\pm$ 0.66% | - | - |
| Qwen 3-32B-Sympy | - | 48.00% $\pm$ 4.62% | - |
| Qwen 3-32B-Sphinx | - | - | 35.61% $\pm$ 1.31% |
| GLM-4.5-Air | 58.58% $\pm$ 1.39% | 56.00% $\pm$ 1.33% | 48.87% $\pm$ 1.98% |
| Devstral-Small-2-24B | **62.63%** $\pm$ **1.32%** | **56.24%** $\pm$ **3.27%** | **53.79%** $\pm$ **4.73%** |

# D. Truncation Ratio Curve

We order $T_1$ trajectories from **Sera-4.5A-Lite** based on the ratio of trajectory steps that fit in 32768 tokens, a property we term "truncation ratio". We partition the ordered $T_1$ trajectories into subsets of 3000 samples each. This forces each subsequent partition to contain trajectories with strictly lower truncation ratios than the previous partition. We then train Qwen3-32B on each partition. $T_1$ trajectories work well because they are longer than $T_2$ trajectories on average while exhibiting similar scaling trends. This allows us to study the effect of training on a wide range of truncation ratios with a non-trivial amount of data and expect findings to translate.

In Figure 6 we plot SWE-Bench Verified performance against the average truncation ratio from every partition. Surprisingly, we find that the best data comes from trajectories that have high truncation ratios but are not fully contained in 32768 tokens. Subsequent truncation ratios result in gradually decreasing performance. We suspect that this is due to a combination of factors, such as longer trajectories reflecting more difficult tasks and that a model's final steps are typically focused on the redundant task of submitting its solution instead of problem solving.

# E. Cost Breakdown

We assume a cost of $2 per H100 GPU-hour throughout this section, which reflects current cloud pricing for on-demand instances.

**Reinforcement learning.** RL-based approaches for coding agents require substantial compute. SkyRL-Agent (Cao et al., 2025) reports 4,601 H100-hours to train SA-SWE-32B, yielding a cost of $9,202 and achieving 39.4% on SWE-bench Verified. For comparison, DeepSWE (Luo et al., 2025) requires 9,180 H100-hours ($18,360) to reach similar performance. **SERA**'s total cost for data generation and training is 960 H100-hours ($1,920), making it 4.8$\times$ cheaper than SkyRL in raw compute and 9.6$\times$ cheaper than DeepSWE. However, **SERA** also achieves higher data efficiency. We fit a power law to **SERA**'s cost-performance curve (Figure 1) and find that **SERA** reaches SkyRL's 39.4% at a cost of just $352 when self-hosting via vLLM, or $173 via the z.ai API. This yields a cost-to-performance efficiency of 26$\times$ (vLLM) or 53$\times$ (z.ai) compared to SkyRL.

**Synthetic data generation.** Figure 1 shows scaling curves under self-hosted inference via vLLM. We also calculate costs for API-based inference using GLM-4.5-Air and GLM-4.6 through the z.ai API, which we find to be the cheapest. To derive the API cost, we analyzed 100 randomly sampled trajectories from the SWE-smith trajectory dataset (Yang et al., 2025) to measure actual token consumption patterns. Each trajectory consists of multiple API calls where the conversation history

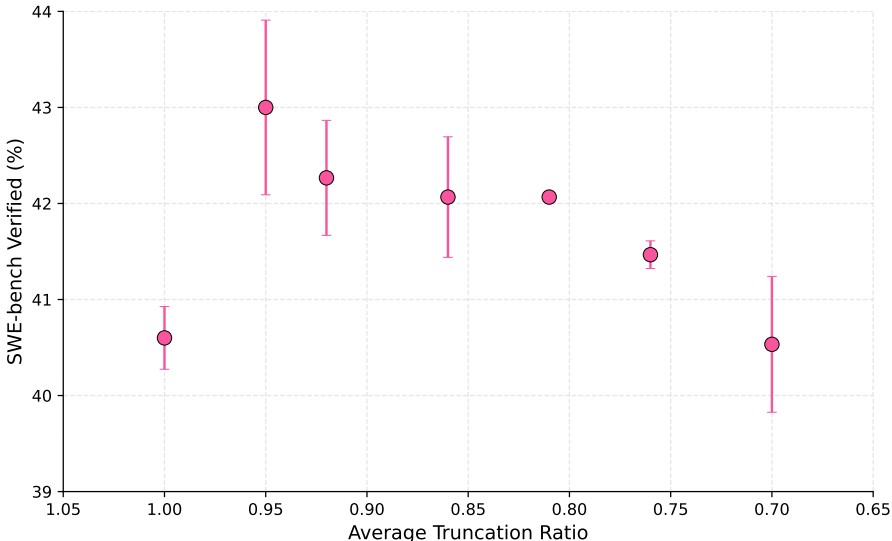

*Figure 6.* SWE-bench Verified performance vs. truncation ratio at 32K context. Each point represents 3,000 $\mathbf{T}_1$ trajectories partitioned by truncation ratio, averaged over 3 seeds. Trajectories with truncation ratio 0.95 perform best.

grows with each turn. For a given API call, the model receives the full conversation history (cached input), the new tool result or observation (uncached input), and produces a response (output). We measured these components and rescaled to 32K context length to match our training setup, yielding an average of 35 API calls per trajectory.

Table 9 shows the per-trajectory cost breakdown across four configurations: SWE-smith using the Sonnet 3.7 API, **SERA** using the z.ai API with GLM-4.5-Air and GLM-4.6, and **SERA** self-hosted via vLLM. For the API-based methods, we show the token-level breakdown; for vLLM, we report the GPU cost directly.

*Table 8.* API pricing used for cost calculations.

| Provider | Input (/MTok) | Cached (/MTok) | Output (/MTok) |
|---|---|---|---|
| Anthropic (Sonnet 3.7) | $3.00 | $0.30 | $15.00 |
| z.ai (GLM-4.5-Air) | $0.20 | $0.03 | $1.10 |
| z.ai (GLM-4.6) | $0.60 | $0.11 | $2.20 |
| vLLM (self-hosted) | 0.065 GPU-hours/trajectory × $2/GPU-hour | | |

*Table 9.* Cost breakdown per trajectory. Token cost percentages show the share of total billed tokens across 35 API calls per trajectory, rescaled to 32K context. See Table 8 for pricing details.

| Component | % Token Cost | SWE-smith (Sonnet 3.7) | SERA (GLM-4.5-Air) | SERA (GLM-4.6) | SERA (vLLM) |
|---|---|---|---|---|---|
| Cached input (context) | 95.9% | $0.2247 | $0.0225 | $0.0824 | — |
| New input (tool results) | 3.1% | $0.0730 | $0.0049 | $0.0146 | — |
| Output (generations) | 1.0% | $0.1151 | $0.0084 | $0.0169 | — |
| Issue creation | — | $0.0540 | — | — | — |
| **Inference subtotal** | — | $0.4668 | $0.0358 | $0.1139 | $0.1307 |
| Training | — | $0.0560 | $0.0560 | $0.0560 | $0.0560 |
| **Total per trajectory** | — | **$0.5228** | **$0.0918** | **$0.1699** | **$0.1867** |

The dominant cost for API-based methods is the cached conversation context, which accumulates approximately 749K tokens across 35 API calls per trajectory. For Sonnet 3.7, even with prompt caching at $0.30/MTok, the cumulative context

accounts for 54.4% of inference cost. Output tokens, though far fewer (7.7K per trajectory), are disproportionately expensive due to the higher output price ($15.00/MTok). SWE-smith additionally requires $0.054 per trajectory for synthetic issue creation. In total, **SERA** with GLM-4.5-Air via the z.ai API is $5.7\times$ cheaper than SWE-smith with Sonnet 3.7, and $2.0\times$ cheaper than self-hosting via vLLM. GLM-4.6 via the z.ai API costs $0.1699 per trajectory—$3.1\times$ cheaper than SWE-smith and $1.1\times$ cheaper than vLLM self-hosting, while providing a stronger teacher model. The $3.2\times$ higher inference cost of GLM-4.6 compared to GLM-4.5-Air ($0.1139 vs $0.0358) is partially offset by the fixed training cost ($0.056), yielding only a $1.85\times$ increase in total per-trajectory cost. Importantly, these per-trajectory comparisons do not account for data quality. As shown in Table 2, **SERA** achieves higher performance per sample than competing methods, and GLM-4.6 produces higher-quality data than GLM-4.5-Air at comparable sample sizes. When we account for this by comparing the cost to reach equivalent performance levels using our scaling law, the effective advantages are substantially larger: **SERA** reaches SWE-smith's 32.6% (Qwen 2.5) performance at a cost of $60 (vLLM) or $29 (z.ai with GLM-4.5-Air), compared to SWE-smith's $3,395. This yields a cost-to-performance efficiency of $57\times$ (vLLM) or $115\times$ (z.ai with GLM-4.5-Air).

At scale, **SERA** requires approximately $1.5K to generate 16,000 trajectories via the z.ai API with GLM-4.5-Air, $2.7K with GLM-4.6, compared to $3.0K via vLLM and $8.4K via the Sonnet 3.7 API. The scaling law in Figure 1 predicts that with GLM-4.6 via the z.ai API, matching Devstral-Small-2 performance requires approximately $6K in data generation cost, compared to $23K with GLM-4.5-Air via the z.ai API and $47K with vLLM self-hosting.

However, we note important caveats for using commercial APIs in research. API pricing is subject to change, and providers may adjust model quality, rate limits, or availability without notice. This makes experiments difficult to reproduce exactly and can introduce confounding factors if model behavior shifts between experimental runs. For these reasons, APIs may not be suitable for rigorous scientific work that demands full reproducibility. We still encourage researchers to consider the vLLM backend with open-weight models, which provides complete control over the inference process and ensures consistent behavior across experimental runs. That said, for practitioners operating under cost constraints who need to generate training data quickly, a commercial API with cached input pricing offers a viable alternative at substantially reduced cost.

## F. Data Filtering for Specialization

**Long Edit and Tool Call Length Filtering:** Table 10 highlights the effect of filtering out trajectories with long edits and large tool calls during specialization. We classify a excessively long edit as final edits exceeding 40 lines and excessively large tool calls as tool call responses containing more than 600 tokens. Ablating these filtering conditions for each repository, we find that no single filter setting reliably generalizes for all repositories. Indeed, filtering long edits works very well for Django and Sympy, but is ineffective for Sphinx, which instead benefits from filtering for tool call size.

*Table 10.* Effect of filtering on repository-specialized data. "Patch $\leq$40 Lines" drops trajectories with patches exceeding 40 lines. "Both Filters" additionally removes trajectories where average tool output exceeds 600 tokens. Filtering patches improves Django (+2.3%) and Sympy (+4.4%), while Sphinx benefits from the combined filter (+4.9%). Trained on Qwen 3-32B, evaluated at 32K context. Results averaged over 3 seeds.

| Repository | No Filter | Patch $\leq$40 Lines | Both Filters |
|---|---|---|---|
| Django | $49.93\% \pm 1.64\%$ | $\mathbf{52.23\% \pm 1.64\%}$ | $50.07\% \pm 2.95\%$ |
| Sympy | $46.67\% \pm 2.67\%$ | $\mathbf{51.11\% \pm 1.54\%}$ | $44.89\% \pm 1.54\%$ |
| Sphinx | $32.29\% \pm 1.94\%$ | $30.30\% \pm 6.95\%$ | $\mathbf{37.14\% \pm 6.95\%}$ |

We also apply these filtering techniques to $T_1$ trajectories from **Sera-4.5A-Lite**, as shown in Table 11, which results in no significant improvements.

*Table 11.* Effect of filtering on general $T_1$ trajectories from GLM-4.5-Air. "Patch $\leq$40 Lines" removes trajectories with patches exceeding 40 lines (n=5,364 from n=7,400). "Tool Output $\leq$600" removes trajectories where average tool output exceeds 600 tokens (n=6,136 from n=7,400). Neither filter improves performance on general data. Trained on Qwen 3-32B, evaluated at 32K context. Results averaged over 3 seeds.

| No Filter | Patch $\leq$40 Lines | Tool Output $\leq$600 |
|---|---|---|
| $43.93\% \pm 1.30\%$ | $43.67\% \pm 1.60\%$ | $44.00\% \pm 0.00\%$ |

Taken together, our results suggest that filtering can be targeted to improve performance on specific repositories but is not as reliable in aggregate. The effectiveness of filtering methods likely reflects individual codebase characteristics. As a result, we suggest that users develop their own filtering heuristics for personal repositories.

**Specializing to Multiple Repositories:** We also train on Django and Sympy jointly to investigate whether SERA can be used to specialize to multiple codebases at once. We randomly sample half our initial Django and Sympy datasets from Section 4.3, training on the combined dataset. We jointly evaluate Django and Sympy instances separately, then average them with equal weighting, which debiases against the larger number of Django instances in SWE-Bench Verified. We find that while performance drops slightly on each constituent codebase, the average performance of the combined dataset outperforms 10000 $T_1$ trajectories from **Sera-4.5A-Lite** (Table 12). In addition, independent performances on Django and Sympy still compare favorably to the teacher model. This indicates that **SERA** can be applied to multiple codebases for broadly improved performance, which reflects the needs of enterprises and larger research teams.

*Table 12.* Multi-repository specialization results. Specialized training on 8,000 single-repository trajectories achieves the best per-repository performance. Mixed training (4,000 Django + 4,000 Sympy) achieves the best average. General training uses 10,000 $T_1$ trajectories from **Sera-4.5A-Lite**. Average is computed with equal weighting between Django and Sympy. Trained on Qwen 3-32B, evaluated at 32K context. Results averaged over 3 seeds.

| Training Data | Django | Sympy | Average |
|---|---|---|---|
| Specialized (8k Django) | **52.23% ± 1.64%** | — | — |
| Specialized (8k Sympy) | — | **51.11% ± 1.54%** | — |
| Specialized (4k Django + 4k Sympy) | 50.07% ± 0.50% | 46.22% ± 3.35% | **48.15%** |
| General (10k samples) | 45.60% ± 0.90% | 48.89% ± 0.77% | 47.25% |

# G. Teacher Models

In Appendix B, we show that GLM-4.5-Air is a much better teacher than Claude 3.7 Sonnet despite similar SWE-Bench Verified performance. We hypothesize that this is in part due to GLM-4.5-Air's reasoning traces, which are longer and significantly more elaborate.

To study the effect of reasoning traces, we train on 4200 $T_2$ trajectories from GLM-4.5-Air where we remove reasoning traces and leave only tool calls. In Table 13 we find that this significantly degrades performance compared to the unchanged trajectories. Our results confirm our hypothesis that high-quality reasoning traces are essential when distilling data for coding agents.

*Table 13.* Effect of reasoning traces on coding agent performance. Both conditions use 4,200 $T_2$ trajectories from GLM-4.5-Air. No Reasoning removes all reasoning traces, retaining only tool calls. Trained on Qwen 3-32B, evaluated at 32K context. With Reasoning results averaged over 3 seeds.

| Condition | SWE-bench Verified |
|---|---|
| With Reasoning | **41.00% ± 1.31%** |
| No Reasoning | 23.00% |

# H. Rollout Mixing

In Section 4.3, we mix $T_1$ and $T_2$ trajectories to increase sample size during specialization. We repeat this experiment at a significantly larger scale in Table 14 using GLM-4.6 as a teacher. Combining 16,000 $T_2$ trajectories and 9,224 $T_1$ trajectories improves performance compared to training only on 16,000 $T_2$ trajectories. While the resulting model falls just shy of scaling only $T_2$ trajectories, the results suggest that $T_1$ and $T_2$ can be reliably mixed to extract further performance gains in data constrained settings.

Assumptions of ANOVA are also met. A one-way ANOVA reveals no statistically significant difference between trajectory mixing strategies, $F(2,6) = 1.91$, $p = .229$, $\eta^2 = .39$. This indicates that $T_1$ and $T_2$ trajectories can be combined without degrading performance, enabling improved sampling efficiency.

*Table 14.* Effect of mixing $\mathbf{T_1}$ and $\mathbf{T_2}$ trajectories. All data generated using GLM-4.6 as teacher. 16k $\mathbf{T_2}$ uses 16,000 second-rollout trajectories. 25k $\mathbf{T_2}$ uses 25,224 second-rollout trajectories. Mixed condition combines 16,000 $\mathbf{T_2}$ with 9,224 $\mathbf{T_1}$ trajectories. Mixing improves over 16k $\mathbf{T_2}$ alone (+1.9%) but falls slightly short of scaling to 25k $\mathbf{T_2}$. Trained on Qwen 3-32B, evaluated at 32K context. Results averaged over 3 seeds.

|  | 16k $\mathbf{T_2}$ | 25k $\mathbf{T_2}$ | 16k $\mathbf{T_2}$ + 9k $\mathbf{T_1}$ |
|---|---|---|---|
| SWE-bench Verified | 47.07% $\pm$ 1.21% | **49.53% $\pm$ 1.94%** | 49.00% $\pm$ 1.64% |

## I. Robustness of Evaluations

To assess the reliability of our findings, we conducted a systematic statistical analysis across all experiments in this paper. Our analysis aggregates within and between experiments and for multiple random seeds that include all experiments for scaling laws, verification thresholds, truncation strategies, specialization mixtures, filtering ablations, and baseline comparisons. In total, this analysis covers 78 experimental conditions, each evaluated with three random seeds, yielding 234 individual evaluation runs. Based on our findings we concluded with recommended best practices for coding agent evaluations.

**Observed Variance:** Across all experimental conditions, we observe standard deviations ranging from 0.5% to 3.0%, with a median of 1.2%. This is problematic when the magnitude of improvement in coding agent research is typically also 1–3%. Many reported gains in the literature fall within one standard deviation of run-to-run noise.

**Signal-to-Noise Analysis:** A practical way to assess whether an observed improvement is real is to compute the signal-to-noise ratio (SNR): the absolute difference between methods divided by the typical run-to-run variance. When SNR < 1, noise dominates and the result cannot be trusted. When SNR is between 1–2, the result is borderline and requires more seeds. When SNR > 2, there is likely a real effect. Applying this framework to our experiments:

- **High confidence (SNR > 3):** Specialized vs. general data (+4.3%, SNR=5.6), SERA vs. SWE-smith with same teacher (+4.7%, SNR=4.4), scaling law predictions (mean error 0.4%)
- **Moderate confidence (SNR 2–3):** Verification threshold equivalence (all within 2.9%, SNR confirms no difference), truncation ratio effects (+2.4%, SNR=2.2)
- **Low confidence (SNR < 2):** Student matching teacher at 8k samples (1.7% difference, SNR=1.4, error bars overlap)

**How Many Seeds Do You Need?** Based on the empirical variance in our data (median standard deviation of 1.2%), Table 15 shows approximately how many seeds are required to achieve SNR $\geq$ 2 for different effect sizes. These estimates follow directly from the definition: to achieve SNR = 2 for an effect of size $\delta$, the standard error must be at most $\delta/2$, requiring $n \approx (2 \cdot \text{std}/\delta)^2$ seeds.

*Table 15.* Seeds required to achieve SNR $\geq$ 2 for different effect sizes, derived from the empirical variance in our experiments (median std = 1.2%). With only 3 seeds, improvements below 2–3% should be treated with skepticism.

| Effect Size | Seeds for SNR $\geq$ 2 | Reliability with $n = 3$ |
|---|---|---|
| 1% | $\sim$15 | Cannot detect reliably |
| 2% | $\sim$4 | Borderline |
| 3% | $\sim$2 | Adequate |
| 5% | $\sim$2 | High confidence |

**The Single-Seed Problem:** Many published results in coding agent research report single-seed evaluations. Our data demonstrates the danger of this practice. Across multiple experiments, we find cases where different random seeds lead to opposite conclusions about which method is best. For example, in our truncation experiments, seeds 1 and 2 identify ratio 0.95 as optimal, while seed 3 identifies ratio 0.92 as optimal with 0.95 performing 2.2% worse. Single-seed ablations cannot be trusted.

**Cross-Model Generalization:** An concerning observation is that methods might not generalize well across different base models or teacher models. We observe that SWE-smith achieves 32.6% with Qwen 2.5-32B but only 25.3% with Qwen 3-32B. This 7.3% drop suggesting the method may have been unintentionally optimized for the earlier model family. For

our method, changing the teacher model to Sonnet 3.7 and Sonnet 4.0 behaves as expected, demonstrating cross-model generalization. However, we did not have resources to test cross-model generalizations for the base model. Even our findings should be interpreted with caution: improvements we observe may not transfer to base models outside the Qwen and GLM families.

**Scaling Laws as a Robustness Check:** We found scaling laws to be invaluable for ensuring reliable results and recommend that future work in coding agents incorporate them where possible. Our scaling experiments (Figure 1) show that performance follows a highly predictable power law ($R^2 > 0.95$, mean prediction error 0.4%). Scaling laws provide several benefits: (1) *experimentation efficiency*: running experiments at smaller, cheaper scales while extrapolating findings to larger scales, since power laws have proven predictable and reliable; (2) *cost estimation*: predicting the resources required to reach target performance levels before committing to expensive runs; (3) *method comparison*: estimating sample efficiency and cost differences between methods without running exhaustive experiments at all scales; and (4) *robustness checking*: when a method's performance falls significantly outside the scaling law prediction, this signals either a genuine breakthrough or, more likely, noise or overfitting to a particular configuration.

**Recommendations:** Based on this analysis, we encourage researchers to (1) run a minimum of 3 seeds, preferably more for ablations expecting improvements below 3%, (2) report standard deviations alongside means, (3) compute the signal-to-noise ratio and treat SNR $< 2$ results as preliminary, (4) verify that improvements transfer across model configurations, and (5) fit scaling laws where feasible to enable efficient experimentation and robustness checking.

