# OpenReview forum: "SERA: Soft-Verified Efficient Repository Agents"
_ICML.cc/2026/Conference — ICML 2026 regular_

### Official Review · Reviewer_rgKk · 2026-02-13

**Soundness:** 3
**Presentation:** 4
**Significance:** 3
**Originality:** 4
**Overall Recommendation:** 5
**Confidence:** 5

**Summary:**

The authors present Soft Verified Generation (SVG), a synthetic teacher-model-based approach to generate thousands of trajectories from a single code repository. Unlike previous synthetic data approaches (like SWE-smith), their approach relies purely on inference from a teacher model via multiple rollouts, in a bug-prompt based injection to yield a representative trajectory, yielding a synthetic PR, which then the teach is tasked with solving, generating a trajectory and a patch that is soft-verified against the original patch. Using this method, they collect a large corpus of data for supervised finetuning (SFT) and train an 8B/32B Qwen model to very competitive performance, even relative to some similarly sized closed and open model recipes.

Most interestingly, they conduct a great deal of ablations to validate their choice of teacher, context length, scaling in general, and particularly repository specific specialization scaling laws, where they show that SVG and the consequent SERA models, when specialized on a repo, can actually outperform the teacher model (Figure 3). These additional ablations show some interesting findings, namely (Section 5.1) showing that verification is not needed for high quality data (i.e. model’s subtasks are also very useful learning signal even if the E2E task is not solved).

**Compliance With Llm Reviewing Policy:**

Affirmed.

**Final Justification:**

Authors address my main concerns, I maintain my score of clear accept (5).

**Key Questions For Authors:**

(see above)

**Limitations:**

yes

**Strengths And Weaknesses:**

Strengths:

* Comprehensiveness of evaluation:
  * All the experiments in this work are extremely thorough and well founded. I appreciated the clarify of the diagrams and direct comparisons to both strong and comparable models (Table 1 alone is a huge contribution).
  * Each of the ablations is also well motivated and gives the reader a comprehensive comparison to prior work and shows exactly where SERA/SVG’s improvements are coming from (teacher choice, data filtering choice, etc.)
* Strong presentation, significance to the field, and paper narrative:
  * The paper’s narrative and motivation are extremely timely and well-founded, especially with the huge recent interest in coding agents. I also highly appreciate how the authors motivated the need for open-code models against closed source models, explaining how closed source models cannot hope to specialize to private or personal codebases.  I also appreciate the Claude Code integration :)
* Cost effectiveness of method relative to prior work:
  * The authors show very strong experimental results for the cost-effectiveness of code agent post-training, both not requiring RL and also being significantly cost wise cheaper than previous synthetic data methods. Especially, since many code post-training methods price out most academics, this is an extremely valuable finding for the community.

Weaknesses:

As a clear disclaimer to the AC, in comparison to the strengths, these are much closer to nits and “nice-to-haves,” but I would love to hear the authors clarify these points, as (in my mind) they still prevent me from seeing such a method as Pareto better than other synthetic/RL methods.

* Clarity of method explanations:
  * The exact details of SVG are still unclear to me (Line 127):
  * Is the bug prompt inserting a bug into the codebase or is it just enough for us to explore and get a nice trajectory demonstrating agentic skills from a teacher model M? I think this section would benefit from more detail on exactly what is in each of the colored units in Lines 140 to 146\.
    * In other words, is this first rollout done on functions where there are no bugs (and the intentional vagueness yields us a nice trajectory)?
    * If this is the case (in tandem with the results showing that trajectories are valid regardless of level of verification), is SERA’s improvement primarily due to this being a more optimal way to distill teacher agentic characteristics into a student model? The authors mention this briefly in the limitations on saturation, but would love to hear more here.
* SVG and specialization on fast moving repositories:
  * I wonder if such a method requires frequent updates as specialized model knowledge gets out of sync, especially in the case of fast moving repositories. Did you observe if using SVG results in specialized models memorizing possibly outdated features of repositories? One concern I had from such an approach was if models specialized on a repo are robust to large drastic changes in that repo.
    * Alternatively, were the repo specific finetunes tested again on the wider SWE-bench Verified evaluation?

---

> ### Author Rebuttal · Authors · 2026-03-31
>
> **Q1: Is the bug prompt inserting a bug into the codebase or is it just enough for us to explore and get a nice trajectory demonstrating agentic skills from a teacher model M? I think this section would benefit from more detail on exactly what is in each of the colored units in Lines 140 to 146. In other words, is this first rollout done on functions where there are no bugs (and the intentional vagueness yields us a nice trajectory)? If this is the case (in tandem with the results showing that trajectories are valid regardless of level of verification), is SERA’s improvement primarily due to this being a more optimal way to distill teacher agentic characteristics into a student model?**
>
> **Method Clarification:**
>
> In detail, SVG works as follows:
>
> Rollout 1:
>
> Given a codebase $C$ in its initial state, we ask the teacher to fix a bug in the codebase, vaguely guiding it with a randomly sampled bug description $bug_j$ (example included below). The prompt includes a random function $func_i$ to start from, ensuring the teacher doesn’t collapse to the same change every time. So yes - the first rollouts are done on functions without any deliberately introduced bugs.
>
> We have the teacher self evaluate its change against the given bug description, to make sure its change is aligned with the prompt. Once the self evaluation passes, we create the synthetic PR describing the patch $P_1$ that was just produced.
>
> Rollout 2:
>
> The goal of rollout 2 is to produce the desired change described by $synth_{PR}$. In other words, to replicate the chance that was just created. In the second rollout, we again provide the teacher with the original codebase but now just prompt it with $synth_{PR}$.
> We compare the patch $P_2$ from this rollout against the patch from Rollout 1 to evaluate how well the model was able to reproduce $P_1$.
>
> The important intuition for SVG is that we believe coding agent tasks are better described as trying to produce a desired change instead of necessarily fixing a unit-test breaking bug.
>
> **Why it Works:**
>
> We agree with the reviewer’s intuition. Our results suggest that the reasoning & skills demonstrated during rollouts could be more important than final correctness. This hypothesis is also supported by other recent works in distillation [1] and past work in k-shot learning [2]. Following our submission, we added ~22,000 existing trajectories from Rollout 1 into our training dataset for SERA-32B, increasing from 25000 to 47000 samples. Training on this, performance increases from 49.5% to 50.7%, outperforming not just open data models but also leading open-weight models like Devstral-Small-2 and even Qwen3.5-27B (49% @ 32k) at 32K context.
>
> [1] https://arxiv.org/pdf/2512.22255
> [2] https://arxiv.org/abs/2202.12837
>
> **Q2: Did you observe if using SVG results in specialized models memorizing possibly outdated features of repositories? Alternatively, were the repo specific finetunes tested again on the wider SWE-bench Verified evaluation?**
>
> We investigate this and do not find evidence that outdated repository features are being memorized. Our experiment is as follows:
>
> We take our Django specialization dataset and subsample 1500 trajectories from 3 Django commits before 12/29/2021 – representing half of the 6 Django commits in the specialization dataset. The six Django commits in the original specialization dataset span the entire Django evaluation timespan in SWE-Bench Verified.
>
> We then evaluate all test instances after 12/29/2021, totalling 69 instances. We bin these instances into three bins, simulating low, medium, and high repository drift over time. Each bin contains 23 instances.
>
> | Django ID Range        | Time Range |  Time Limited Model  (1500 samples)   | Non-Time Limited Model (1500 samples) |
> |--------------|------------|-----------------|-----------------|
> | 15252–15741  |     Dec 29 2021 - Jul 2 2022    | 34.77% ± 7.56%   |        44.90% ± 5.02%         |
> | 15814–16493  |     Jul 2 2022 - Jan 19 2023     | 50.73% ± 2.54%   |         47.80% ± 0.00%        |
> | 16502–17087  |     Jan 19 2023 - Jul 17 2023   |  50.70% ± 13.29%  |         52.17% ± 4.35%        |
>
> Interestingly, the most significant performance difference between the two training settings occurs with test instances with the smallest time drift. As time drift increases, performance equalizes. This suggests that SVG does not just memorize snapshotted features. One explanation for this is that training on multiple, earlier commits (three from < 12/29/2021) encourages the model to prioritize learning stable repository features that are unlikely to change.
>
> Still, drastic changes may confuse a specialized model. For these reasons, we believe recent works in test time training are complementary to our work and look to incorporate these strategies in ongoing follow up work.

---

> > ### Author Rebuttal · Reviewer_rgKk · 2026-04-01
> >
> > I thank the authors for their responses and comprehensive experimentation, and believe my concerns to be fully addressed. I am definitely very impressed by the rigor and clarity of this paper, and, together with the comprehensive rebuttal, the depth of thinking behind each of the experiments/ablations.
> >
> > The only wider concern I have is that such a method feels somewhat close to training/data methods that were built to fix the "reversal curse" in early LLMs (i.e. inserting a bug with context then reconstructing, is similar to flipping if A, then B), but this is a wider concern on synthetic training data approahces and not specific to this paper. The increase in metrics is certainly impressive alone and has important learnings for folks training coding agents.
> >
> > I maintain my score of clear accept (5).

---

> > > ### Author Response · Authors · 2026-04-03
> > >
> > > Thank you for your feedback and kind words!
> > >
> > > We understand your concerns with synthetic data, but we believe that when done right, it has enormous potential to create large-scale datasets targeted towards both general and specific tasks; all while being affordable in a way real data is not.

---

### Official Review · Reviewer_dec6 · 2026-03-12

**Soundness:** 4
**Presentation:** 2
**Significance:** 3
**Originality:** 4
**Overall Recommendation:** 5
**Confidence:** 5

**Summary:**

This work trains language models for agentic SWE tasks using synthetically generated data. The key contribution is a data generation approach that is more lightweight than prior bug-specific approaches and can provide similar performance. Training data is collected from an agent tasked with modifying a code base and soft-verification is performed using line-matching metrics. The author's leverage the lightweight nature of their approach to scale data generation and train high performing models. They also use this approach for training repository specific models.

**Compliance With Llm Reviewing Policy:**

Affirmed.

**Final Justification:**

The rebuttal addressed minor concerns I had. Overall, I think this is a strong paper that should be accepted.

**Key Questions For Authors:**

1. Would training on only T1 and T2 trajectories have been sufficient? Is soft verification even necessary if all you need is agent interaction data?
2. What is the SOTA performance figure claimed in Sec 4.2?
3. Would data generated in this way help improve performance on repositories that the base LLM has never seen before (private / proprietary code)?

**Limitations:**

Yes

**Strengths And Weaknesses:**

Demonstrating a data generation approach that doesn't require test harnesses can perform on par with prior more expensive approaches is an a very useful contribution. This has downstream applications to training agents in a much cheaper way and also for training repo-specific models. The approach is conceptually simple and evaluations are thorough, providing sufficient evidence for the central claims. I recommend this paper be accepted.

Weaknesses:
- The paper does not do a good job of addressing why their specific soft-verification approach works. Would training on only T1 and T2 trajectories have been sufficient? Is soft verification even necessary if all you need is agent interaction data?
- The ablation in section 5.1 claims to understand whether unit tests verification is necessary. But the experiments performed are on differing values of line matching threshold. This does not directly link to unit tests which test semantic correctness of code through execution.
- Some points in the paper are confusingly written.
    - Section 3.1 should be clearer about what exactly is the content of things like T, synth_PR. Maybe the prompts could be provided in the appendix.
    - Section 4.2 claims that scaling the data results in SOTA performance. A table with numbers to support this would help.
- The repository specialization experiments are performed on popular python repositories which are well represented in llm pre-training data. It is unclear whether this approach will generalise to truly private repositories (which is pitched as a possible benefit).
- While this approach is useful for distillation, it does not address how frontier models themselves may get better at coding through interaction.

---

> ### Author Rebuttal · Authors · 2026-03-31
>
> **Q1: Would training on only T1 and T2 trajectories be sufficient? Is soft verification necessary if all you need is agent interaction data?**
>
> A1: We believe it is too strong of a claim to say that verification is not needed at all. However, we our method works because even incorrect rollouts can contain useful training signals, which concurrent works have also noted [1]. The patterns learned from unverified T1 and T2 may eventually saturate, at which point higher verification thresholds from soft-verification may be required.
>
> To answer the reviewer’s question about T1 and T2 trajectories, we combine T1 and T2 trajectories from GLM-4.6 into a larger dataset of 47000 trajectories. We train Qwen3-32B on this set and observe that performance continues to increase, reaching 50.7% +/- 1.9% on SWE-Bench Verified. This is SoTA performance among all <= 32B open-weight models at 32K context – outperforming Devstral Small 2 and the recent Qwen3.5-27B (49% @ 32K). Consequently, unverified T1 and T2 data remains effective up to the largest scales we test.
>
> [1] https://arxiv.org/pdf/2512.22255
>
> **Q2: The experiments performed are on differing values of line matching threshold. This does not directly link to unit tests which test semantic correctness of code.**
>
> A2: We respectfully disagree that there is no link between soft verification and semantic correctness of code. We conduct two experiments that indicate the opposite is true.
>
> In the first, we aggregate every SWE-Bench Verified evaluation we ran over the last nine months (total of 1036). For every evaluation, we calculate the average soft verification score against the gold patch and average unit test pass rate. Our evaluations span numerous models trained by us in the last year and also other model families (Claude, GPT, Devstral, etc.). In this setup, across 1036 runs we observe an extremely strong correlation between soft verification and test pass rate, with r=0.898. This is visualized in Figure 1 of the anonymous link.
>
> In our second experiment, we aim to remove bias within runs. To do this, we calculate the soft verification score for every SWE-Bench Verified instance across all runs, totalling 516,997 datapoints. Next, we separate the scores into 20 bins from [0, 0.05] … [0.95, 1.00]. We calculate the average unit test success in each bin. The results, shown in Figure 2, again demonstrate a remarkable correlation between soft verification and unit test success.
>
> Our results indicate that there is a strong link between soft verification and unit tests, with soft verification having the added benefit of being cheaper, quicker to run, and generalizable to any codebase.
>
> Results Link: https://sera-rebuttal-plots.tiiny.site/
>
> **Q3: Clarify T, synth_PR. Maybe prompts could be provided in the appendix.**
>
> A3: Thank you for the feedback. $T$ is the generated multi-turn trajectory, alternating between user and assistant roles. $synth_{PR}$ is the synthetic PR that is generated using Rollout 1 and then fed into Rollout 2 as the task prompt. We will update the appendix with the examples. While examples will not fit in this response due to character constraints, we are happy to provide them in the discussion period.
>
> **Q4: SoTA Results Table**
>
> A4: We show these results in Table 1 and will make it more clear in the paper.
>
> **Q5: It is unclear whether this approach will generalise to truly private repositories (which is pitched as a benefit).**
>
> A5: We agree with the reviewer that this is a limitation. We choose to demonstrate specialization on Django, Sympy, and Sphinx because they have validated evaluation instances that can be used to measure accuracy. That being said, we are actively working on accessing & testing on unseen codebases in follow up work. In the meantime, we will update the paper to be more clear about this limitation.
>
> **Q6: Approach does not address how frontier models may get better at coding through interaction.**
>
> A6: We respectfully disagree that this is a weakness. Reinforcement learning and real data are important at the frontier, but our focus is on smaller, specialized coding agents. In this setting, distillation is much more cost efficient than alternative approaches. We derive these savings in Appendix E, finding that our approach requires 4.8x less GPU hours than SkyRL and 9.6x less GPU hours than DeepSWE to reach the same performance.
>
> **Q7: Would data generated this way help perf on repos that the base LLM has never seen (private / proprietary code)?**
>
> A7: Yes, we believe that is one of the contributions of our work. Right now, most training data for coding agents is limited to repositories with unit tests, which create implicit biases for codebases with higher code quality. But many personal codebases lack unit tests and are less documented, less modular, etc. Data generated agnostic to unit tests will allow data to be generated in these long tail repositories, which can improve coding agent performance in downstream applications.

---

> > ### Author Rebuttal · Reviewer_dec6 · 2026-04-03
> >
> > Thank you for your thorough response. I believe the work is a strong contribution as I mentioned in my original review and maintain that it should be accepted.
> >
> > The findings you presented in the rebuttal about correlation between soft-verification and unit-test results are especially interesting and would be great to include in the paper.
> >
> > Your result in how using combined T1 and T2 trajectories achieves a SOTA model is also very interesting. I believe it is evidence to say that while soft-verification can help filter good data, using unfitered data directly is ultimately the best approach for someone who wants to distill a larger model into a smaller model? While this does not detract form the contributions of the paper, it seems like an important implication that should be be discussed.
> >
> > For your response to Q7, do you have evidence to support this?

---

> > > ### Author Response · Authors · 2026-04-05
> > >
> > > Yes - when done right,  unfiltered data appears best for distillation because not only is there no observable quality drop-off, but it also increases the number of samples available for training. We agree that this is important to discuss in more depth and will update the paper accordingly.
> > >
> > > While we do not have direct experiments supporting the hypothesis in Q7, we plan to investigate it in follow up work. We believe it is a reasonable hypothesis given what we know about synthetic data curation in literature. Current real and synthetic data approaches rely on converting unit tests from failing to passing to validate each instance. Unit test agnostic approaches -- of which SVG is the first -- avoid biases stemming from this requirement while adding diversity into the train set, creating substantial room for performance gains.
> > >
> > > We appreciate your engagement throughout this review cycle. Please consider raising your score if we have alleviated your concerns.

---

### Official Review · Reviewer_GtfP · 2026-03-13

**Soundness:** 3
**Presentation:** 3
**Significance:** 4
**Originality:** 3
**Overall Recommendation:** 4
**Confidence:** 4

**Summary:**

This paper proposes Soft-Verified Efficient Repository Agents (SERA), a framework for rapidly and cost-effectively producing repository-specialized agents tailored to private codebases. With supervised fine-tuning (SFT) alone, SERA achieves strong performance among fully open-source models (open data, methods, and code) and is competitive with open-weight baselines such as Devstral-Small-2.The authors report that constructing SERA incurs substantially lower training cost than reinforcement-learning-based approaches, and that SVG can generate thousands of trajectories from a single repository. Moreover, the paper demonstrates that repository specialization on individual projects (e.g., Django, SymPy, and Sphinx) can match or even surpass the teacher model with relatively small amounts of synthetic data.

**Compliance With Llm Reviewing Policy:**

Affirmed.

**Key Questions For Authors:**

1、Is the demonstration PR sampled from the SWE-bench Verified evaluation split? If so, how do you ensure strict non-overlap with evaluation instances? Could you instead use SWE-bench (train) or other publicly available PRs as demonstrations and reproduce the key results?
2、Beyond SWE-bench Verified, have you validated the same trends (e.g., the limited importance of verification strength) on other agent benchmarks or evaluation settings?

**Limitations:**

Yes

**Strengths And Weaknesses:**

Strengths：
1、Strong engineering and research value; substantially lowered training barrier. The approach avoids reliance on an executable test environment, bug injection, or complex reinforcement-learning (RL) infrastructure, which makes repository specialization—particularly for private codebases—more practical and deployable. This is potentially impactful for both the open-source community and industrial applications.
2、The paper reports that SERA-32B attains performance close to, or on par with, strong open-weight baselines on SWE-bench Verified, and remains competitive under both 32K and 64K context settings. The authors also highlight important evaluation considerations such as context-length effects, fairness in comparisons, and cache-related issues, which improves the credibility of the experimental results.
3、Findings such as verification strength may not be crucial, truncation ratio substantially affects data quality, and mixing rollouts is beneficial provide useful guidance for future work on synthetic data generation and training repository agents.
Weaknesses：
1、The authors show that, at a certain data scale, different soft-verification thresholds (including r = 0) yield similar downstream performance, and conclude that verification may not matter. While this may hold in the reported regime, a concern is that such data could primarily teach tool use, code retrieval, and editing procedures, rather than semantic correctness or test-passing reliability. Moreover, in higher-performance regimes (near saturation) or on more challenging tasks, correctness signals may become important again (a possibility the paper briefly acknowledges in its limitations).
I would like to see (i) a small-scale unit-test-based validation on a subset of runnable repositories to estimate the correlation between soft verification scores and actual test pass rates, and/or (ii) evidence that this conclusion is robust across broader benchmarks, not only SWE-bench Verified.
2、In the second step of SVG, the pipeline samples a demonstration pull request (PR) from SWE-bench Verified to teach the teacher model the PR-writing format. Even if the intent is purely to learn formatting, this design introduces a non-trivial risk of leaking evaluation-set content into the generation process—particularly if the sampled PRs are drawn from the same distribution as the evaluation instances, or potentially overlap with them.

---

> ### Author Rebuttal · Authors · 2026-03-31
>
> **Q1: Would like (i) a small-scale unit-test-based validation on a subset of runnable repositories to estimate correlation between soft verification scores and actual test pass rates, and/or (ii) evidence that this conclusion is robust across broader benchmarks, not only SWE-bench Verified.**
>
> **A1.1:** While soft verification is not equivalent to unit test pass rates, we find a strong correlation between the two. We conduct two experiments that support this.
>
> In the first, we aggregate every SWE-Bench Verified evaluation we ran over the last nine months (total of 1036). For every evaluation, we calculate the average soft verification score against the gold patch and average unit test pass rate. Our evaluations span numerous models trained by us in the last year and also other model families (Claude, GPT, Devstral, etc.). In this setup, across 1036 runs we observe an extremely strong correlation between soft verification and test pass rate, with r=0.898. This is visualized in Figure 1 of the anonymous link.
>
> In our second experiment, we aim to remove bias within runs. To do this, we calculate the soft verification score for every SWE-Bench Verified instance across all runs, totalling 516, 997 datapoints. Next, we separate the scores into 20 bins from [0, 0.05] … [0.95, 1.00]. We calculate the average unit test success in each bin. The results, shown in Figure 2, again demonstrate a remarkable correlation between soft verification and unit test success.
>
> Results Link: https://sera-rebuttal-plots.tiiny.site/
>
> **A1.2:** Agent training has many confounding variables, which is a huge reason progress has been slow. So we decided to focus on SWE-bench Verified and prioritized using our compute to eliminate as many confounding variables as possible. While this focus makes the paper less general, we believe it strengthens this work as a community contribution. We devise a lightweight data pipeline and conduct numerous data experiments that others can build on across numerous scales, so overall agent training can be studied with less compute. We hope this will accelerate more general insights across many other datasets over time.
>
> **Q2: In the second step of SVG, the pipeline samples a demonstration pull request (PR) from SWE-bench Verified to teach the teacher model the PR-writing format. Even if the intent is purely to learn formatting, this design introduces a non-trivial risk of leaking evaluation-set content into the generation process—particularly if the sampled PRs are drawn from the same distribution as the evaluation instances, or potentially overlap with them.**
>
> We conduct experiments on the format of demonstration issues comparing the use of 1) SWE-Bench Verified issues and 2) a standardized, unseen PR format. We find no difference in performance between the two settings, suggesting that there is no evaluation leakage occurring.
>
> In more detail, we prompt Claude to create an unseen, general PR format (shown below), which we use as a demonstration instead of a SWE-Bench issue. In this setup, SWE-Bench issues are never exposed at any point in the point. We generated and trained on 1347 T2 trajectories in this setting using GLM-4.5-Air as the teacher. In this setting, we found no significant difference against 1500 T2 trajectories using SWE-Bench issues as demonstrations (as in the paper).
>
> | SWE-Bench Verified (%) | Unseen Issue | SWE-Bench Issue |
> |------------|------------:|------------:|
> | Average | 38.27 | 38.87 |
> | Std. Dev. | 1.10 | 1.15 |
> | Train Samples | 1347 | 1500 |
>
> This suggested that using SWE-Bench issues as demonstrations doesn’t introduce contamination or performance bias.
>
> In our paper, we decide to mirror SWE-smith and use SWE-Bench issues as demonstrations to ensure that demonstration PRs are not confounding variables in head to head comparisons (Table 2).
>
> General PR format:
> ```
> metadata:
>   id: <unique-identifier>
>   type: <bug|feature|documentation|performance|security|refactor|test>
>
> project:
>   repository: <repository-url>
>   component: <affected-component-or-module>
>   version: <version-where-issue-occurs>
>
> title: <concise-descriptive-title>
>
> description:
>   summary: |
>     <brief-overview-of-the-issue>
>
> reproduction:
>   preconditions: |
>     <setup-or-state-required-before-reproduction>
>
>   steps: |
>     <numbered-steps-to-reproduce>
>
>   code_sample: |
>     <minimal-reproducible-code-example>
>
>   input_data: |
>     <sample-data-if-applicable>
>
> behavior:
>   actual: |
>     <what-currently-happens>
>
>   expected: |
>     <what-should-happen>
> ```
>
> **Q3: Is the demonstration PR sampled from the SWE-bench Verified evaluation? If so, how do you ensure non-overlap with evaluation instances? Could you instead use SWE-bench (train) or other publicly available PRs as demonstrations and reproduce key results?**
>
> See A2.
>
> **Q4: Beyond SWE-bench Verified, have you validated the same trends (e.g., the limited importance of verification strength) on other agent benchmarks or evaluation settings?**
>
> See A1.2

---

> > ### Author Rebuttal · Reviewer_GtfP · 2026-04-03
> >
> > The response has well addressed my comments.

---

> > > ### Author Response · Authors · 2026-04-03
> > >
> > > Thank you for your feedback! Please consider raising your score if we have alleviated your concerns.

---

### Official Review · Reviewer_AVzF · 2026-03-15

**Soundness:** 3
**Presentation:** 3
**Significance:** 3
**Originality:** 3
**Overall Recommendation:** 5
**Confidence:** 4

**Summary:**

This paper presents Soft-Verified Efficient Repository Agents (SERA), a method for training open-weight coding agents that are both high-performing and cost-effective. The core innovation is Soft Verified Generation (SVG), a pipeline that generates synthetic training data without the need for complex bug-injection or expensive test-suite infrastructure. By using soft verification, comparing generated patches through line-level recall rather than execution. the authors significantly lower the barrier to specializing agents for private codebases.

**Compliance With Llm Reviewing Policy:**

Affirmed.

**Final Justification:**

The authors have addressed my concerns, and therefore I am changing my overall recommendation to Accept.

**Key Questions For Authors:**

1. The authors noted that a student can outperform a teacher through repository specialization. At what point does the "quality" of the teacher's reasoning traces become a hard ceiling that repository-specific knowledge can no longer overcome?
2. The current experiments utilize codebases that are primarily Python-based. How does the soft verification (line-level recall) performance hold up in languages with more rigid syntax or different structural paradigms (e.g., C++ or Rust)?
3. Since unverified trajectories are used for training, is there a risk that the model learns hallucinated tool-use patterns that might pass soft verification but fail in real-world environments where a compiler is present?

**Limitations:**

Yes

**Strengths And Weaknesses:**

**Strengths**

- SERA achieves the best results among fully open-source models on SWE-bench Verified, matching or exceeding much larger or closed-source models like Devstral-Small-2.
- SERA is 26x cheaper than RL and 57x cheaper than previous synthetic methods to reach equivalent performance.
- The method allows models to be fine-tuned on specific private repositories, where a student model can eventually outperform its teacher by encoding repository-specific knowledge directly into its weights.
- The authors provide a detailed statistical analysis of the noise in coding agent benchmarks, recommending best practices like using at least three seeds and scaling laws for robustness.


**Weaknesses**

- While soft verification works well for initial skill gains, the authors admit that reinforcement learning or hard verification may eventually be necessary once a model saturates on basic codebase navigation.
- The study focuses almost exclusively on SWE-bench Verified, which may limit the understanding of how well these agents generalize to broader, non-Python tasks.
- Filtering techniques for data quality (like patch length) appear to be codebase-specific; a filter that helps on Django might hurt on Sphinx, requiring users to develop their own heuristics.

---

> ### Author Rebuttal · Authors · 2026-03-31
>
> **Q1: While soft verification works for initial gains, authors admit reinforcement learning or hard verification may eventually be necessary once a model saturates on basic codebase navigation.**
>
> While we are careful with claims in the paper, all evidence points to strong scaling of our method without any slowing. To provide further evidence, we scaled to even larger training runs reaching SoTA for all models <=32B. Furthermore, we computed correlations between unit test pass rate and soft-verification to provide further evidence that our method scales if verification would be important at scale.
>
> Scaling from 24k to 47k samples improved SERA 32B further to 50.7% +/- 1.9%, yielding SoTA performance that exceeds all comparable open models when controlling for context length -- including the new Qwen 3.5 27B model (49% @32k). This suggests that the reasoning & skills demonstrated during rollouts could be more important than final correctness. This hypothesis is also supported by other recent work in distillation [1] and past work in k-shot learning [2].
>
> Correlations between soft-verification and test pass rate are r=0.898 over 516,997 instances from 1036 SWE-Bench runs – a near perfect correlation. This means that soft-verification can be seen as a continuous proxy for hard verification. If scaling depends on verification, our method can be used in a straightforward manner by using a larger verification threshold over time.
>
> Detailed correlation figures can be found in Figure 1 and 2 at this anonymized link: https://sera-rebuttal-plots.tiiny.site.
>
> [1] https://arxiv.org/abs/2512.22255
> [2] https://arxiv.org/abs/2202.12837
>
> **Q2: When does "quality" of the teacher's reasoning traces become a ceiling that repository-specific knowledge cannot overcome?**
>
> We observe that scaling our method led to exceeding the teacher in all cases. This is commonly observed in distillation literature. However, we also observed that this can be inefficient and one should switch to a larger model long before a teacher is saturated or exceeded. All this is consistent with the distillation literature.
>
> That said, we conduct a best estimate of ceilings using our general GLM-4.5-Air scaling curve in Figure 1. Between 7400 samples and 16000 samples from the general scaling law, average performance increases from 44.00% to 46.60% on SWE-Bench Verified before plateauing – a 1.06x increase.
>
> During repository specialization, we train for 8000 trajectories from each codebase. Assuming similar scaling trends apply, this suggests that performance will be bounded as follows.
>
> | Row Title | Reported % | 1.06x Increase | Plateau % |
> |-----------|------------|----------------|-------------|
> | Django | 52.2 | 3.09 | 55.29 |
> | Sympy | 51.1 | 3.02 | 54.12 |
> | Sphinx | 37.1 | 2.19 | 39.29 |
>
> **Q3: The study focuses on SWE-bench Verified, which may limit understanding of how these agents generalize to broader, non-Python tasks.**
>
> Agent training has many confounding variables, which is a huge reason progress has been slow. So we decided to focus on SWE-bench Verified and prioritized using our compute to eliminate as many confounding variables as possible. While this focus makes the paper less general, we believe it strengthens this work as a community contribution. We devise a lightweight data pipeline and conduct numerous data experiments that others can build on across numerous scales, so overall agent training can be studied with less compute. We hope this will accelerate more general insights across many other datasets over time.
>
> **Q4: Filtering techniques for data (eg patch length) appear codebase-specific; a filter that helps Django might hurt Sphinx, requiring users to develop their own heuristics.**
>
> The related work FrogBoss tests this directly by producing verified synthetic data that has long, multi-file patches. We found that sample-by-sample our data improves SWE-bench performance as much as FrogBoss does (see results in appendix). This indicates that patch length is only weakly related to data quality. We use patch length filtering not for data quality purposes but primarily to be able to generate soft-verified data at diverse thresholds since complex patches naturally skew the distribution.
>
> **Q5: Soft verification performance in languages w/rigid syntax or different structural paradigms (e.g., C++ or Rust)**
>
> See A3.
>
> **Q6: Is there a risk the model learns hallucinated tool patterns that pass soft verification but fail in real-world environments?**
>
> We do not see any evidence of this. We observe tool call issues only when switching scaffolds outside of the training distribution. For example, we train with SWE-agent so we notice new failures when deploying SERA (SWE-agent) in Claude Code, which uses a different scaffold. We handle this by translating tool calls from SWE-agent to the Claude Code tool call format. This seems to eliminate all tool-call problems.

---

> > ### Author Rebuttal · Reviewer_AVzF · 2026-04-03
> >
> > Authors addressed all concerns raised.

---

> > > ### Author Response · Authors · 2026-04-03
> > >
> > > Thank you for your feedback! Please consider raising your score if we have alleviated your concerns.

---

### Decision · Program_Chairs · 2026-04-30

**Decision:**

Accept (regular)

**Comment:**

This paper proposes SERA, a soft verified generation framework that addresses the prohibitive cost of execution-based verification in coding agent training data. By pragmatically trading absolute correctness for scalability, the paper achieves a strong cost-performance balance.

The reviewers are unanimously positive. The core strength of this work lies in its practical impact: it delivers a simple yet scalable pipeline that makes open-source models competitive while drastically cutting training complexity. The empirical section is robust, earning high praise for its controlled comparisons, scaling analysis, and repository-specialization studies.

While the reliance on soft verification and the exclusive focus on SWE-bench Verified represent clear limitations, the rebuttal successfully mitigated these concerns by clarifying verification quality metrics and dispelling doubts about data leakage. The practical utility of this approach is undeniable, and I fully support accepting the paper.